# Immunohistochemical Expression of TGF-β1 in Kidneys of Cats with Chronic Kidney Disease

**DOI:** 10.3390/vetsci9030114

**Published:** 2022-03-03

**Authors:** Yuki Uehara, Yu Furusawa, Md Shafiqul Islam, Osamu Yamato, Hitoshi Hatai, Osamu Ichii, Akira Yabuki

**Affiliations:** 1Laboratory of Veterinary Clinical Pathology, Joint Faculty of Veterinary Medicine, Kagoshima University, 1-21-24 Korimoto, Kagoshima 890-0065, Japan; k6991729@kadai.jp (Y.U.); si.mamun@ymail.com (M.S.I.); osam@vet.kagoshima-u.ac.jp (O.Y.); 2Kagoshima University Veterinary Teaching Hospital, Joint Faculty of Veterinary Medicine, Kagoshima University, 1-21-24 Korimoto, Kagoshima 890-0065, Japan; yu_furusawa728@yahoo.co.jp; 3Laboratory of Veterinary Histopathology, Joint Faculty of Veterinary Medicine, 1-21-24 Korimoto, Kagoshima University, Kagoshima 890-0065, Japan; hhti@vet.kagoshima-u.ac.jp; 4Laboratory of Anatomy, Department of Basic Veterinary Sciences, Faculty of Veterinary Medicine, Hokkaido University, Hokkaido 060-0818, Japan; ichi-o@vetmed.hokudai.ac.jp; 5Laboratory of Agrobiomedical Science, Faculty of Agriculture, Hokkaido University, Sapporo 060-8589, Japan

**Keywords:** cat, chronic kidney disease, immunohistochemistry, renal expression, transforming growth factor-β1

## Abstract

Transforming growth factor-beta 1 (TGF-β1) plays a central role in the progression of chronic kidney disease (CKD). However, in feline CKD, renal expression of TGF-β1 and how it changes as the disease progresses have not been fully studied. In the present study, we immunohistochemically assessed the renal expression levels of TGF-β1 in cats with CKD and statistically analyzed its correlation with CKD severity. Clear immunosignals were detected in the glomerular mesangial cells, Bowman’s capsules, proximal tubules, distal nephrons, platelets, and vascular smooth muscles in the kidneys of cats with CKD. Statistically, luminal signals in the distal nephrons showed positive correlations with plasma creatinine levels and glomerulosclerosis, while those in the proximal tubules and platelets showed negative correlations with plasma urea and/or creatinine levels. Therefore, it was suggested that the changes in the renal expression of TGF-β1 could be associated with progression of feline CKD.

## 1. Introduction

Chronic kidney disease (CKD) is characterized by persistent abnormalities in kidney structure and function. CKD has become a major health problem worldwide, affecting both humans and animals. It is one of the most common diseases in older animals, especially cats, and has become more prevalent in recent decades [1,2,3]. It is induced by various factors, including genetic defects, infections, inflammatory and immune-mediated diseases, and exposure to certain drugs and toxins. Despite diverse underlying causes, CKD ultimately progresses to tubulointerstitial fibrosis (TIF) via a common pathway in humans and cats [4,5,6,7,8,9].

Transforming growth factor-beta 1 (TGF-β1) is a cytokine that is critical for the progression of CKD and its upregulation can lead to progressive renal fibrosis [2,10]. High urinary TGF-β1 expression has been reported in cats with CKD [11,12,13]. Moreover, the transcript level of TGF-β1 in the kidneys is elevated in naturally occurring CKD [14]. However, the localization of TGF-β1 in renal tissues and the changes in its levels in CKD have not yet been investigated in cats. In this study, we analyzed the renal expression levels of TGF-β1 in cats with CKD using immunohistochemistry and statistically evaluated the relationship between the expression of TGF-β1 and CKD severity.

## 2. Materials and Methods

Kidney samples from cats were collected during post-mortem examinations at Kagoshima University, Kagoshima, Japan. CKD was diagnosed based on medical records and renal histopathology. Patients with persistently and stably high plasma creatinine (pCre) concentrations (>140 mmol/L) and low urinary specific gravity (USG) (<1.035) were diagnosed with CKD, but those with large fluctuations in pCre levels were excluded from the study. Cats with a normal range of pCre concentration but that showed renal contraction, persistent low USG, and tubulointerstitial fibrosis in histopathological observations were regarded as having early-stage CKD. Patients with acute kidney injury (AKI), acute-on-CKD, polycystic kidney disease, or neoplastic kidney disease were excluded from this study. Moreover, cases in which glomerulonephritis (membranous or membranoproliferative), pyonephritis, or tubular necrosis were found during histopathological examinations were also excluded from this study. We examined the formalin-fixed and paraffin-embedded (FFPE) blocks of kidney tissues from 41 cats with kidney disease prior to the study. Of these, 12 blocks of kidney tissue from cats with CKD were selected for the present study based on the above criteria. The median pCre concentration in these 12 cats was 185 mmol/L (range 79.6–274 mmol/L). The median age of six cats was 14.5 years (range, 10–17 years), but the ages of the other cats were unknown. In addition, FFPE blocks of kidney tissue from four normal cats (intact males, 23–27 months old) were obtained from Hokkaido University, Hokkaido, Japan and used as normal controls. These cats were purchased from Kitayama Labes Co., Ltd., Nagano, Japan, and used as normal controls for another study that adhered to the AAALAC International standards.

Then, 3 µm thick tissue sections were cut from the FFPE blocks and stained with hematoxylin-eosin, periodic acid-Schiff, and Masson’s trichrome stain for histopathological analysis. Immunohistochemistry was used to detect the TGF-β1 immunosignals. After dewaxing the sections, antigen retrieval was performed through microwave heating in 10 mM citrate buffer (pH 6.0). The sections were treated with 3% hydrogen peroxide (H_2_O_2_) and blocked with 0.25% casein (Sigma-Aldrich, St. Louis, MO, USA). The sections were then incubated with anti-TGF-β1 rabbit polyclonal antibody (1:200; ARP37894; Aviva Systems Biology, San Diego, CA, USA) overnight at 4 °C. This antibody was expected to cross-react with the tissues from the cats based on its immunogen sequence, and its specificity was confirmed using Western blot analysis. After washing with physiological buffered saline, the sections were incubated with a peroxidase-polymer-conjugated universal antibody (Simple Stain MAX-PO (MULTI); Nichirei Biosciences, Tokyo, Japan) at room temperature for 30 min. Immunoreactivity was determined using 3,3’-diaminobenzidine (Merck, Darmstadt, Germany). Counterstaining was performed using Mayer’s hematoxylin solution. For negative control tissue sections, the anti-TGF-β1 antibody pre-incubated with the TGF-β peptide (AAP37894; Aviva Systems Biology) and normal rabbit IgG antibody (Thermo Fisher Scientific, Waltham, MA, USA) was used instead of the primary antibody.

Immunosignals for TGF-β1 were evaluated quantitatively. In this analysis, information about the cases was masked so that the observer could not recognize individual cases. Under light microscopy (×200 magnification), the number of observation fields containing TGF-β1-positive tubules, positive-infiltrated cells, and positive blood vessels were counted, and the positive targets or fields/all fields were calculated for each case (20–50 fields/each case). The tubules of the distal nephrons were evaluated separately in those with positive-luminal space and in those with positive-apical surfaces. Blood vessels were evaluated separately in those with positive media and in those with positive platelets in the lumen. The number of glomeruli with positive Bowman’s capsules and those with positive mesangial cells were counted, and each number was divided by the number of all observed glomeruli. Sections stained with Masson’s trichrome were used for semiquantitative analysis of glomerulosclerosis and TIF, and the analysis was performed according to a previously described method [7]. Spearman’s correlation coefficient was used to evaluate the correlations among the various parameters. Statistical analysis was performed using the freely available software EZR ver. 1.54 (Saitama Medical Center, Jichi Medical University, Saitama, Japan) [15], which is a modified version of R commander designed to add statistical functions frequently used in biostatistics.

## 3. Results

In normal kidneys, minor and faint labeling for TGF-β1 was observed in the external walls of some Bowman’s capsules, a few platelets in the blood vessels, and some vascular smooth muscles (Figure 1).

In CKD kidneys, strong immunosignals for TGF-β1 were detected in the apical and luminal surfaces of the distal nephrons (distal tubules and/or collecting ducts) in all cases (Figure 2a,b). Moreover, significant positive correlations were observed between the TGF-β1 score of the luminal surface of the distal nephron and pCre levels and the glomerulosclerosis score (Table 1). The proximal tubules also exhibited immunolabeling of TGF-β1 in CKD kidneys. Signals were observed in the cytoplasm (Figure 2c). Quantitative analysis revealed a significant negative correlation between the TGF-β1 score of the proximal tubules and pCre levels (Table 1). Immunosignals for TGF-β1 were also detected in the external walls of the Bowman’s capsule, glomerular mesangial cells, platelets in the blood vessels, and vascular smooth muscles (Figure 2d,e). Immunolabeling was also observed in the interstitial infiltrated cells; however, there were very few positively labeled cells (Figure 2f). Among these signals, the TGF-β1 score of the platelets showed significant negative correlations with pCre levels and plasma concentrations of urea (Table 1).

## 4. Discussion

An increase in urinary TGF-β1 levels is known as one of the pathological events in human CKD [16,17]. In cats, recent studies have also shown that urinary excretion of TGF-β1 is increased in CKD [11,12,13]. However, feeding a commercially available, phosphate-restricted, renal diet to cases with CKD produced no significant changes of urinary TGF-β1 excretion [18]. Moreover, an increase in the transcription level of TGF-β1 in the kidneys has been reported in naturally occurring feline CKD [19]. The present study showed significant positive correlations between the TGF-β1 score of the luminal surface of the distal nephrons and pCre levels and the glomerulosclerosis score. Based on these findings, we suggest that the urinary excretion of TGF-β1 in feline CKD may be caused by renal tissue expression and glomerular hyperfiltration of TGF-β1. As the progression of glomerulosclerosis induces hyperfiltration in the residual glomeruli [20], it could also induce the hyperfiltration of TGF-β1 in CKD. Therefore, the filtered TGF-β1 could be condensed in the distal nephrons and excreted via urine. In addition, it is possible that TGF-β1 expression in the distal nephron might be induced by glomerulosclerosis. Expression of TGF-β1 in the distal nephron has been previously demonstrated in an in vitro study using Madin–Darby canine kidney (MDCK) cells [21]. In that study, urinary albumin was demonstrated to be an inducible factor for the upregulation of TGF-β in MDCK cells. Proteinuria (albuminuria) is a principal symptom of glomerulosclerosis and is caused by podocyte injury in sclerotic glomeruli [22,23]. As proteinuria is a well-known prognostic factor and therapeutic target for feline CKD [24,25,26], proteinuria caused by glomerulosclerosis might be an inducible factor of TGF-β1 expression in the distal nephron of cats. However, no direct evidence could be presented due to the lack of urinalysis data for these cases.

In the present study, the TGF-β1 score of the proximal tubules showed a negative correlation with pCre levels. TGF-β1 is thought to play a primary role in the progression of tissue fibrosis, and upregulation of TGF-β1-driven signaling induces pro-fibrotic effects in interstitial fibrosis in CKD [6]. In a recent study using isolated and cultured feline proximal tubular epithelial cells (FPTEC), it was demonstrated that the exposure of FPTEC to TGF-β1 induced loss of epithelial morphology and alterations in gene expression consistent with the occurrence of the partial epithelial–mesenchymal transition (EMT) [19]. Another recent study using cultured feline kidney epithelial cells also demonstrated the profibrotic effect of TGF-β1, which induces the expression of the EMT markers such as α-smooth muscle actin, collagen type 1, and others [27]. EMT is a phenomenon in which epithelial cells acquire mesenchymal traits and is one of the most important events for induction of interstitial fibrosis in CKD, and the occurrence of EMT in proximal tubules in feline CKD has already been suggested in our previous immunohistochemical study [9]. These reports, together with the present findings, suggest that the expression of TGF-β1 in the proximal tubules is associated with the early pathogenesis of feline CKD and may act as a pathomolecular trigger event that induces the incidence and progression of interstitial fibrosis. In addition, we inferred that immunosignals for TGF-β1 in the proximal tubules might reflect regenerative action to repair damaged tubules. In humans, the proximal tubule is the main target site for AKI. Cytokines, including TGF-β1, promote the regeneration of proximal tubules [28]. Although the cases with AKI and tubular necrosis (a feature of AKI) were excluded from this study, invisible acute tubular injury could potentially occur in the early stages of CKD. It is conceivable that TGF-β1 may be expressed in acutely damaged proximal tubules, which might act as a pathomolecular trigger event that induces the incidence and progression of feline CKD.

TGF-β1 in platelets showed negative correlations with pCre levels and plasma urea. As immunosignals of platelets were weak in the normal kidneys, the present findings might reflect the enhancement of TGF-β1 expression in platelets during the early stages of CKD. High levels of TGF-β1 are found in circulating platelet α-granules, and the activation of these platelets may induce the release of TGF-β1 [29]. Platelet activation is a well-known pathophysiological event associated with renal failure. Tryptophan-derived uremic toxins induce platelet hyperactivity, thereby causing uremic thrombosis [1]. Therefore, the levels of TGF-β1 in platelets might be decreased during the progressive stage of feline CKD due to the excessive release of TGF-β1 from platelets activated by uremic toxins. In addition, TGF-β1 released from activated platelets contributes to liver, cardiac, and aortic fibrosis, most likely by initiating profibrotic signaling and collagen synthesis pathways [30,31,32]. Therefore, it is possible that TGF-β1 released from activated platelets may contribute to TIF, leading to the progression of feline CKD. However, the present study could not demonstrate a significant correlation between the TGF-β1 score of platelets and the severity of TIF.

## 5. Conclusions

In the present study, we immunohistochemically examined changes in renal TGF-β1 expression in feline CKD. The immunosignals of TGF-β1 were observed at various sites in the renal tissue of cats with CKD. The expression levels of TGF-β1 in the distal nephrons might be related to the progression of glomerulosclerosis in CKD, while those in the proximal tubules might contribute to tubular damage, especially in the early stages of CKD. Moreover, platelets can decrease TGF-β1 content as CKD progresses. However, the relationship between the renal expression of TGF-β1 and the severity of TIF was not demonstrated in this study. There are several limitations in the present study. The sample size was small, and most samples were collected from cats at a relatively early stage of stable CKD, as it was difficult to collect samples from those with a later stage of stable CKD during the autopsy. The pCre measurement was not always performed after fasting. The control cats were significantly younger than the cats with CKD, and it was not clear whether age-dependent changes in renal TGF-β1 expression might be related to the pathogenesis of CKD. Complete exclusion of the concomitant pathologies in particular hyperthyroidism was not conducted for all cases. Therefore, further investigation is required to clarify the specific pathophysiological roles of TGF-β1 in feline CKD.

## Figures and Tables

**Figure 1 vetsci-09-00114-f001:**
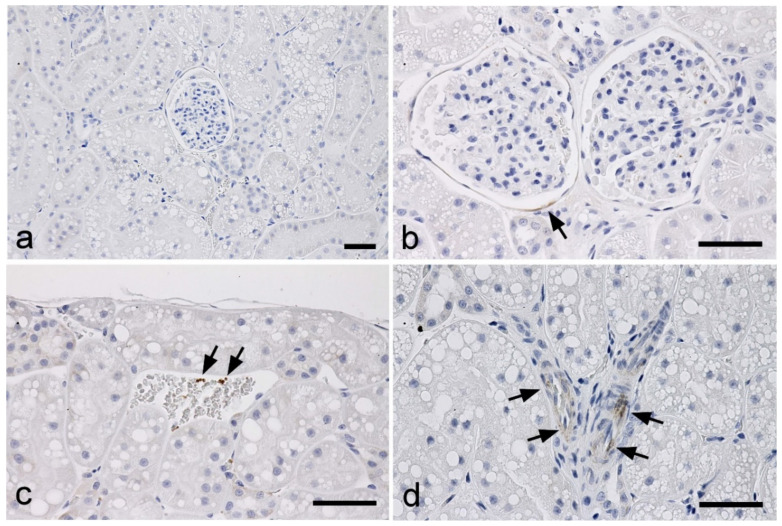
Immunohistochemistry for the detection of TGF-β1 in the kidneys of normal cats. (**a**) No immunolabeling is observed in the cortex. (**b**) A portion of the external walls of Bowman’s capsules showing faint signals (arrow). (**c**) A few platelets of the vascular system showing positive signals (arrows). (**d**) Some vascular smooth muscles showing weak positive signals (arrows). Scale bar: 50 μm.

**Figure 2 vetsci-09-00114-f002:**
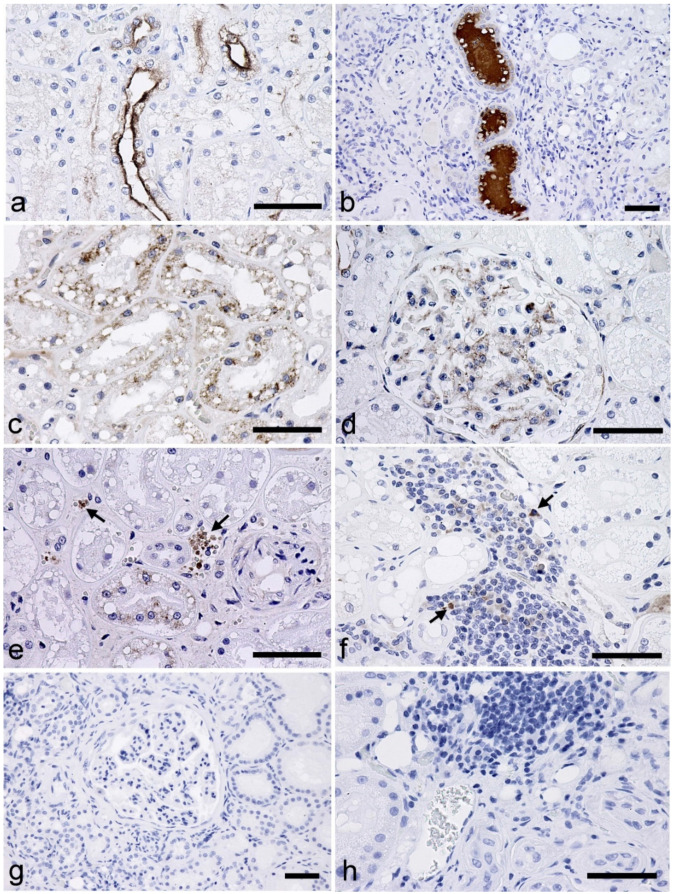
Immunohistochemistry for the detection of the transforming growth factor-beta 1 (TGF-β1) in the kidneys of cats with chronic kidney disease (CKD). (**a**) Immunolabeling is evident in the apical surfaces of the distal tubules and collecting ducts. (**b**) The luminal spaces of some tubules exhibiting strong immunosignals. These tubules are classified as distal tubules or collecting ducts based on their morphologies. (**c**) Granular immunosignals in the proximal tubular cells. (**d**) Immunolabeling in the glomerular mesangial cells. (**e**) Immunosignals in the platelets of the vascular system (arrows). (**f**) Positive cells observed in the focally infiltrated mononuclear cells (arrows). (**g**,**h**) No immunolabeling was observed in the negative control sections. Scale bar: 50 μm.

**Table 1 vetsci-09-00114-t001:** Correlations between the expression of the transforming growth factor (TGF)-beta 1 and tissue damage in the kidneys of felines with chronic kidney disease.

TGF-β1-Positive Sites	Urea	Creatinine	GS	TIF
Bowman’s capsules	NS	NS	NS	NS
Glomerular mesangial cells	NS	NS	NS	NS
Proximal tubules	NS	−0.576 *	NS	NS
Luminal space in DN	NS	0.590 *	0.652 *	NS
Apical surface in DN	NS	NS	NS	NS
Platelets in blood vessels	−0.609 *	−0.813 **	NS	NS
Vascular walls	NS	NS	NS	NS
Infiltrated cells	NS	NS	NS	NS

Spearman’s rank correlation coefficient (* *p* < 0.05, ** *p* < 0.01, NS: not significant); DN: distal nephron; GS: glomerular sclerosis; TIF: tubulointerstitial fibrosis.

## Data Availability

The data are available on request from the corresponding author.

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
