# Peer review of "Immunohistochemical Expression of TGF-β1 in Kidneys of Cats with Chronic Kidney Disease"

_vetsci, 2022, doi:10.3390/vetsci9030114_

Round 1

Reviewer 1 Report

The manuscript vetsci-1592039 describes a case of  Immunohistochemical Expression of TGF-β1 in Kidneys of Cats with Chronic Kidney Disease. 
The report is interesting and current.
I accept it in present form for publication.
For future study, I would suggest considering a correlation with the microenvironment and the immune cells that characterize the chronic inflammation and/or whit other markers of chronic renal damage.

Author Response

Thank you for your review of our manuscript. We are pleased to hear that our paper is acceptable. In addition, we would like to say thank you for your helpful comment. It will be very useful when designing our future study regarding feline CKD.

Reviewer 2 Report

It is an interesting study that analyzes the renal expression levels of TGF-β1 in cats with CKD using immunohistochemistry and statistically evaluated the relationship between the expression of TGF-β1 and CKD severity.

The diagnosis of CKD, in patients included in the study, was based solely on persistently and stably high plasma creatinine (pCre) concentrations (>140 mmol/L). Other parameters, particularly urinary specific gravity, are not indicated.  

Creatinine is an imprecise marker of GFR though, it lacks specificity if reference intervals are set low enough to detect early stage disease, but lacks sensitivity if reference intervals are set higher. pCre is affected by lean tissue mass and hydration, creatinine concentrations (and reference intervals) vary between different assays, analysers and laboratories. The exponential relationship between GFR and creatinine means that substantial early declines in GFR may be accompanied by only small changes in creatinine,while in the latter stages of disease large changes in creatinine may reflect only small changes in GFR (Sparkers et al. SFM Consensus Guidelines on the Diagnosis and Management of Feline Chronic Kidney Disease J Feline Med Surg 2016)

In clinical practice feline CKd is diagnosed on the basis of an increased serum creatinine concentration >140 μmol/l (>1.6 mg/dl) together with an inappropriately low USG (<1.035), and evidence that these changes are sustained (over several weeks or months) or with a history suggesting sustained clinical signs consistent with CKd. It is useful to exclude concomitant pathologies in particular hyperthyroidism.Lack of complete clinical pathological data is an important limitation of the study.

The international Renal interest Society (iRiS) has established a CKd staging system  based on the cat’s fasting creatinine concentration. It would be helpful to differentiate patients, based on this staging system.

Author Response

 We appreciate your constructive comments. As for urinary specific gravity (USG), we apologize for the confusion prompted by our inappropriate description in the conclusion. The persistent low USG was confirmed in all cases used in the present study. It had been described in the first submitted article. However, the cut-off value was not credited in the description, and we added the value (<1.035) in the revised text (lines 54-55). As for the exclusion of the concomitant pathologies in particular hyperthyroidism and measurement of the fasting creatinine concentration, we agree with your suggestions. According to your suggestion, we added these two points as the limitations of our study (lines 215-222).

Reviewer 3 Report

In this interesting study, the authors aimed to investigate the immunohistochemical expression of TGF-β1 in Kidneys of 
Cats with CKD. 
The design of this study is well developed and the experiment adequately performed. 
I suggest you cite in the introduction/discussion section the findings of the study "Lawson JS, Syme HM, Wheeler-Jones CPD, Elliott J. Investigation of the transforming growth factor-beta 1 signalling pathway as a possible link between hyperphosphataemia and renal fibrosis in feline chronic kidney disease. Vet J. 2021 Jan;267:105582. doi: 10.1016/j.tvjl.2020.105582. Epub 2020 Nov 28. PMID: 33375963; PMCID: PMC7814380" and the publication "van Beusekom CD, Zimmering TM. Profibrotic effects of angiotensin II and transforming growth factor beta on feline kidney epithelial cells. J Feline Med Surg. 2019 Aug;21(8):780-787. doi: 10.1177/1098612X18805862. Epub 2018 Oct 22. PMID: 30345862; PMCID: PMC6661713". 
I support the publication of this interesting research. 

Author Response

Thank you for your kind comment. Your advice regarding the referenced articles was very useful for improving our manuscript. I checked out these articles and added appropriate sentences in the discussion (lines 147-149[Ref. 18], 175-177 [Ref. 27]).

Round 2

Reviewer 2 Report

The authors have added the required information and the article is interesting and provides useful information on the possible role of the Transforming growth factor-beta 1 (TGF-β1)  in the progression of chronic kidney disease (CKD).